# A Deep Learning Algorithm for Radiographic Measurements of the Hip in Adults—A Reliability and Agreement Study

**DOI:** 10.3390/diagnostics12112597

**Published:** 2022-10-26

**Authors:** Janni Jensen, Ole Graumann, Søren Overgaard, Oke Gerke, Michael Lundemann, Martin Haagen Haubro, Claus Varnum, Lene Bak, Janne Rasmussen, Lone B. Olsen, Benjamin S. B. Rasmussen

**Affiliations:** 1Department of Radiology, Odense University Hospital, 5000 Odense, Denmark; 2Research and Innovation Unit of Radiology, University of Southern Denmark, 5230 Odense, Denmark; 3Open Patient Data Explorative Network, OPEN, Odense University Hospital, 5000 Odense, Denmark; 4Department of Orthopaedic Surgery and Traumatology, Copenhagen University Hospital, Bispebjerg, 2100 Copenhagen, Denmark; 5Department of Clinical Medicine, Faculty of Health and Medical Sciences, University of Copenhagen, 1165 Copenhagen, Denmark; 6Department of Clinical Research, University of Southern Denmark, 5230 Odense, Denmark; 7Department of Nuclear Medicine, Odense University Hospital, 5000 Odense, Denmark; 8Radiobotics, 1263 Copenhagen, Denmark; 9Department of Orthopedic Surgery and Traumatology, Odense University Hospital, 5000 Odense, Denmark; 10Department of Orthopedic Surgery, Lillebaelt Hospital—Vejle, University Hospital of Southern Denmark, 7100 Vejle, Denmark; 11Department of Regional Health Research, University of Southern Denmark, 5230 Odense, Denmark; 12Department of Radiology, Odense University Hospital, 5700 Svendborg, Denmark; 13CAI-X (Centre for Clinical Artificial Intelligence), Odense University Hospital, University of Southern Denmark, 5230 Odense, Denmark

**Keywords:** machine learning, X-ray, radiology, hip dysplasia, radiography

## Abstract

Hip dysplasia (HD) is a frequent cause of hip pain in skeletally mature patients and may lead to osteoarthritis (OA). An accurate and early diagnosis may postpone, reduce or even prevent the onset of OA and ultimately hip arthroplasty at a young age. The overall aim of this study was to assess the reliability of an algorithm, designed to read pelvic anterior-posterior (AP) radiographs and to estimate the agreement between the algorithm and human readers for measuring (i) lateral center edge angle of Wiberg (LCEA) and (ii) Acetabular index angle (AIA). The algorithm was based on deep-learning models developed using a modified U-net architecture and ResNet 34. The newly developed algorithm was found to be highly reliable when identifying the anatomical landmarks used for measuring LCEA and AIA in pelvic radiographs, thus offering highly consistent measurement outputs. The study showed that manual identification of the same landmarks made by five specialist readers were subject to variance and the level of agreement between the algorithm and human readers was consequently poor with mean measured differences from 0.37 to 9.56° for right LCEA measurements. The algorithm displayed the highest agreement with the senior orthopedic surgeon. With further development, the algorithm may be a good alternative to humans when screening for HD.

## 1. Introduction

Joint deformity as seen in the presence of hip dysplasia is a common cause of hip pain in young skeletally mature patients and may lead to osteoarthritis (OA) [1]. Traditionally, first line modality when diagnosing dysplasia of the hip is radiographs, where measurements taken from a standardized anterior-posterior (AP) pelvic radiograph are used to evaluate the anatomical configuration of the pelvis. The lateral center edge angle of Wiberg (LCEA) describes femoral head coverage by the acetabulum and the acetabular index angle (AIA) quantifies the inclination of the acetabular roof [2].

An accurate and early diagnosis may postpone, reduce or even prevent the onset of OA and ultimately hip arthroplasty at a young age [3]. It was previously reported, though, that patients with non-specific hip pain may be left with symptoms for years before the correct diagnosis of hip dysplasia is made, perhaps because the anatomical deformities indicative of hip dysplasia are not routinely reported in all departments of radiology. They found that the correct diagnosis of hip dysplasia was delayed with up to several years (range: 0–204 months) and more than three confrontations (range: 0–11) with the healthcare system [4,5]. The anatomical deformities associated with hip dysplasia may be diagnosed earlier by using artificial intelligence models and algorithms, possibly as a screening tool, which may also have the potential to improve reader variability and workflow.

The process of developing and testing an algorithm for measuring LCEA and AIA has recently been published and results indicated that an automatic measurement model is feasible [6]. Moreover, Fraiwan and colleagues showed the potential of deep transfer learning for detecting developmental dysplasia of the hip in pelvic radiographs of infants [7]. It has also been suggested that AI is useful for detection and classification of hip dysplasia using ultrasound images [8]. Clinical tests of algorithms for measuring hip parameters such as LCEA and AIA in skeletally mature patients are to the best of the authors’ knowledge limited.

The overall purpose of this study was, in a clinical setting, to assess the performance of an algorithm designed to read pelvic AP radiographs of skeletally mature patients. We aimed to assess reliability of the algorithm and agreement between the algorithm and, respectively, orthopedic surgeons, radiologists and a reporting radiographer for measuring LCEA and AIA.

## 2. Materials and Methods

### 2.1. Study Design

In this retrospective study, we used an algorithm trained to identify several specific segments related to hip dysplasia. The algorithm was applied to 78 pelvic radiographs that were consecutively collected from one center. Moreover, two orthopedic surgeons, two radiologists and one reporting radiographer evaluated all images in regard to LCEA, AIA, and the width of both obturator foramen. The study was approved by the Danish National Committee on Health Research Ethics (Project-ID: 2103745) and registered with the regional health authorities (project-ID: 21/22036). The analyses were carried out in concordance with current Guidelines for Reporting Reliability and Agreement Studies [9,10].

### 2.2. Study Population

Anterior-posterior pelvic radiographs of adults referred to diagnostic workup in relation to non-traumatic hip pain at Odense University Hospital were retrospectively identified and collected in a consecutive manner until the desired sample of 78 was achieved (Section 2.6. Statistical analyses). Inclusion criteria were weight-bearing pelvic radiographs of adults (≥18 years). All weight-bearing pelvic radiographs are taken with the legs internally rotated 15°. Exclusion criteria were the presence of arthroplasty or other types of surgical hardware, signs of congenital abnormalities, surgical or fracture sequela. Radiographs that did not include the entirety of the bony pelvis and the proximal femurs were excluded, as were radiographs that did not show the exposure value. Stratified enrolment by sex and age was applied, such that a similar number of males and females above and below the age of 50 years was present in the sample. Ninety-eight pelvic radiographs were screened consecutively, and inclusion were made according to inclusion and exclusion criteria until the desired sample of 78 pelvic radiographs were obtained (Figure 1). The 78 pelvic radiographs were analyzed by the algorithm and read by all human readers.

### 2.3. Anatomic Definitions

All measurements were made in relation to a horizontal reference line adjoining the most inferior points of the ischial tuberosities. The LCEA was defined as the angle between two lines both drawn from the center of the femoral head (CFH), a line perpendicular to the reference line and a line from the CFH to the lateral sourcil of the acetabulum, respectively. The AIA was defined as the angle between a horizontal line from the medial sourcil (medial aspect of the sclerotic acetabular roof) parallel to the reference line and a line connecting the medial and lateral sourcils [11]. Moreover, the foramen obturator index (FOI), an indicator of pelvis rotation, was calculated as the ratio between the widths of the two foramina. The widths of the foramina was measured at the widest point of the foramina, parallel to the reference line (Figure 2) [12].

### 2.4. Algorithm Development and Training

Automatic measurements were extracted from the pelvic radiographs using a newly developed algorithm (RBhip™, Radiobotics, Copenhagen, Denmark). The algorithm was developed using deep-learning and computer vision and trained on more than 2900 pelvic radiographs. The pelvis, including the acetabulum, and the femoral head and neck were independently segmented using a segmentation model with a modified U-Net architecture trained using augmentation with ResNet34 as a backbone [13].

The horizontal reference was established as a line through the most inferior points on the ischial tuberosities. For the LCEA, the circle best encompassing the femoral head was found through a parameterization of a circle fitted to points along the femoral head contour using least squares. The sourcils in both hips were independently segmented to provide the landmarks of the lateral and medial extent of the weight-bearing area. The sourcil segmentation model was trained primarily using annotations from an orthopedic surgeon and optimized to find the lateral extent of the acetabular roof. This point does not necessarily coincide with the lateral acetabular rim [14,15]. For the FOI, the width of each foramina was calculated at 13 equidistant lines parallel to the horizontal reference line. The FOI was established as the maximum width for the right foramina relative to the maximum width of the left foramina. The algorithm flowchart is depicted in Figure 3.

### 2.5. Data Collection

#### 2.5.1. Human Readers

Five readers, two senior and three junior readers, all accustomed to reading and measuring angles and distances related to hip deformities made all the measurements blinded to each other’s results. No clinical information was available to the readers. The senior readers were a musculoskeletal (MSK) radiologist (LB) and a consultant hip surgeon (CV), with 21 and 8 years of experience, respectively. Moreover, a junior MSK radiologist (JR), a junior hip surgeon (MHH) and a reporting radiographer (LBO) (respectively, 3, 5 and 12 years of experience) made all the measurements. To minimize systematic bias, a protocol with definitions of measurements was distributed to each participant prior to the measurement session (Figure 2). In keeping with daily clinical practice, the readers made all measurements digitally in a picture archiving and communication system (GE Healthcare, IL, USA) and recorded the measurements in a database, REDCap (Research Electronic Data Capture). Five of the radiographs were reported three times by all human readers. Additionally, the exposure index was collected as indirect indication of digital image quality, i.e., noise [16].

#### 2.5.2. Algorithm

The algorithm was running as a Software as a service within the hospital firewall. Images were processed by forwarding the AP radiographs directly from the Picture Archiving and Communication System to a secure Digital Imaging and Communications in Medicine destination. The results were returned as a JavaScript Object Notation file and uploaded electronically to the REDCap database. To assess consistency of the algorithm, all radiographs were read twice approximately two weeks apart.

### 2.6. Statistical Analyses

Mean, standard deviation (SD), and range were calculated for all measurements with scatterplots visualizing bivariate associations. Differences between first and second read by the algorithm were presented descriptively by mean, SD, min, max, first (Q1), and third quartile (Q3). Agreement between the algorithm and individual human readers were estimated and illustrated by Bland-Altman (BA) plots with limits of agreement (LoA), bias, and respective 95% confidence intervals (CI). Assuming normality of data, the LoA are estimates of the range within which 95% of all differences between algorithm and human readers will fall. The bias is the mean measured difference between algorithm and human reader [17,18].

Linear mixed effect models were used to assess factors influencing variance in human data by estimating the repeatability coefficient (RC). The RC is a limit below which an estimated 95% of differences between two measurements is expected to fall. Age, sex, FOI, and noise were treated as fixed factors. Patient, reader, and repeated measurements were considered random effects. We derived RCs for (a) Repeatability; the closeness of repeated measurements of the same patient made under similar conditions by the same reader (intrarater variability analysis), and (b) Reproducibility; the closeness of measurements of the same patient made by readers of varying experience in the same measurement setup (interrater variability analysis). The RCs were calculated as 2.77 times the estimated within-subject SD as derived from the mixed effect model [19].

Sample size was calculated based on the procedure proposed by Lu et al. (2016) for sample size assessment of the Bland-Altman method [17,20]. Assuming an SD of 2.1° and a clinical acceptable agreement limit of 5° (LCEA), a sample size of 176 was required to show a similar agreement between algorithm and human readers, with a power of 80% at a significance level of 5%. Since measurements were carried out on both hips and repeated three times on five patients (2 × 5 additional measurements), the required number of patients was 78.

*p*-values < 0.05 were considered statistically significant. The Stata version 16 (StataCorp. 2019, College Station, TX, USA) was used for all statistical analyses.

## 3. Results

The algorithm was not able to read seven of the 78 included images. Therefore, 71 radiographs were read by the algorithm, resulting in a sample with an average age of 50.1 years [range; 18 to 91] consisting of 36 females and 35 males for agreement analyses. All 78 radiographs were reported by the human readers and included in repeatability and reproducibility estimates.

The algorithm proved highly consistent when double reading all measurements, displaying variances between first and second read that were identical or within the range of machine precision (Table 1). Values for all parameters showed a tendency to be higher when measured by humans, particularly the LCEA measurements. The LCEA (right hip) for humans ranged from 25.8 to 35.0° versus 25.4° when measured by the algorithm. Corresponding values for AIA (right hip) ranged from 4.1 to 6.7° for humans versus 4.7° when measured by the algorithm (Table 2). Scatterplots visually depict human measurements over algorithm measurements (Figure 4).

Using the BA LoA analyses, the bias estimate between human readers and algorithm for bilateral LCEA was statistically different from 0, apart from the right LCEA measurements made by the experienced orthopedic surgeon. Mean measured difference between human readers and the algorithm for the right LCEA ranged from 0.37° (95% CI: −0.61 to 1.36) to 9.56° (95% CI: 8.14 to 10.97), for the experienced orthopedic surgeon and the experienced radiologist, respectively. The corresponding values for left LCEA were 3.56° (95% CI: 2.41 to 4.74) and 10.01° (95% CI: 8.37 to 11.82). Bias for AIA measurements displayed a tendency to center around the zero line for all readers ranging from −0.17° (left) to 2.06° (right) as measured by the junior radiologist and the reporting radiographer, respectively (Figure 5 and Figure 6). Bland-Altman inter-observer agreement between the algorithm and individual human readers, including the LoA and 95% CI, are presented for LCEA and AIA in Table 3 and Table 4.

For the mixed effects model, measurements from all 78 cases and the five human readers were included. For LCEA right, the mixed effect model revealed that patient, reader, and repeated measurement variance were 39.46, 11.88, and 7.44, respectively, indicating the between-patient variance to be the prevailing contributor to variance in data. The RC for a repeated measurement of the same patient by the same human reader was 11.69 (2.77 times √17.80), where the RC increased to 15.09 (2.77 times √ (17.80 + 11.88)) when the same measurement was made on the same patient but by a different human reader. As expected, the between-patient variance was dominant across all measurements. The RCs (same patient, same reader) for AIA was 8.28 and 8.22° for right and left hip, respectively. Corresponding RCs for same patient, different reader increased slightly to 8.85 and 8.52°. In Table 5, the estimated variances derived from the mixed effect model including 95% CI are shown. In Table 6, the estimated RCs for between and within reader measurements are presented.

## 4. Discussion

To the best of our knowledge, this is the first study presenting an algorithm that assesses hip dysplasia in adults in a clinical setting. Radiographic evaluation of the pelvis is commonly the first-line approach in patients suspicious of hip dysplasia, where a set of measurements are used to describe anatomy of the pelvis. Several of those measurements are, however, associated with reader variability [11].

We evaluated an algorithm that provided highly consistent measurements of LCEA and AIA. Agreements between algorithm and human readers were associated with subjective variability. Our data showed that particularly for LCEA measurements, the human readers appeared to obtain higher values than the algorithm. If this holds true, the algorithm will identify more people with potential dysplasia based on the LCEA than the human readers. This finding correlates with previous studies reporting that hip dysplasia may be underdiagnosed radiologically and that an accurate diagnosis of hip dysplasia can be delayed, at times, for several years and following more contacts to the healthcare system [4,5,21]. Perhaps reader variance combined with a systematic overestimation of particularly the LCEA may, in part, explain why hip dysplasia is under-diagnosed. As a technical note, the algorithm, including identification of the extent of the lateral sourcil, was trained based on annotations made primarily by an orthopedic surgeon, which could explain the higher correlation between the senior surgeon and the algorithm found in the current study. This, however, seems to be a suitable fit considering it is the surgeon that ultimately decides whether an operation is required.

An inherent limitation in the current study is the lack of a ground truth against which the algorithm and human measurements can be compared. Establishing a ground truth is, however, a common challenge in radiographic measurements. The proposed algorithm in the current study was consistent but accuracy remains to be proven. However, the bias between the algorithm and the five human readers were noted to be similar to the between-reader variability estimated by the mixed effects model. High measurement variability between human readers has previously been reported in regard to radiographic measurements used to assess hip dysplasia. Although the human readers in the current study were presented with a protocol defining the measurements, the repeatability coefficient was high from a clinical perspective, i.e., ranging from 12 to 15° for LCEA measurements. Perhaps a consensus meeting with all human readers, prior to the measuring sessions, could have clarified definition of landmarks further; particularly, the extent of the lateral sourcil may have affected inter-observer variance. Moreover, the human readers electronically drew and manually positioned a best-fit circle for the femoral head in the PACS which is also an inherent limitation that may have contributed to inconsistency in LCEA measurements. Although the BA LoA for the right LCEA was narrow for reader 3, no other systematic differences in measurement variability were uncovered. Hence, educational background or experience cannot explain the variation in data.

In conclusion, the newly developed algorithm based on deep learning offered consistent measurement outputs for LCEA and AIA when reading pelvic radiographs. Manual identification of the same landmarks made by five human readers were subject to variance, and the level of agreement between the algorithm and human readers was consequently poor, although a tendency was revealed that the senior orthopedic surgeon agreed the most with the algorithm, particularly for LCEA right measurements.

### Clinical Implication

We predict this algorithm as a promising tool in the future with the potential to improve measurement consistency. Potentially, with further development, this algorithm can act as a screening tool within departments of radiology and orthopedic surgery, minimizing delayed diagnosis by identifying abnormal findings on pelvic radiographs suspicious of dysplasia. Considering the vast rising bundle of medical imaging, tools to assist radiologist, physicians and reporting radiographers are needed and integrating locally validated algorithms into clinical practice is essential.

## Figures and Tables

**Figure 1 diagnostics-12-02597-f001:**
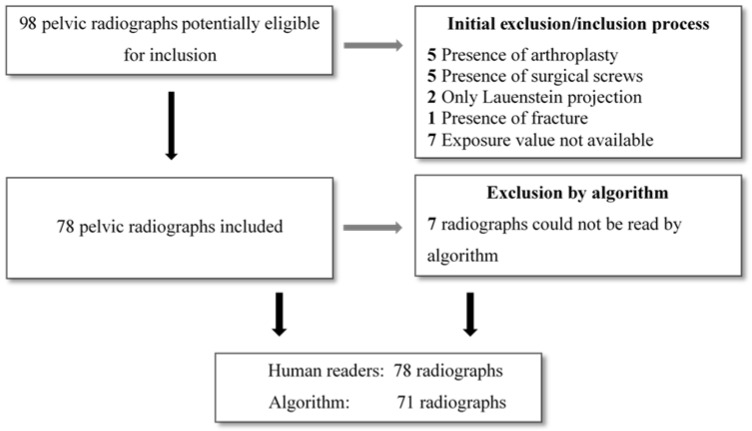
Flow-chart describing the screening process according to inclusion/exclusion criteria. In total, 98 pelvic radiographs were screened until the desired sample of 78 radiographs was achieved.

**Figure 2 diagnostics-12-02597-f002:**
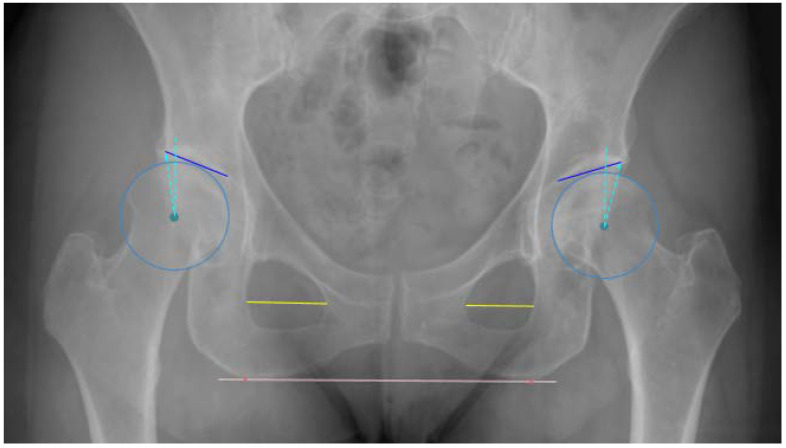
Pelvic radiograph depicting the horizontal reference line (pink line) through the most inferior points on the ischial tuberosities (red dots) and blue circles encompassing both femoral heads. The lateral center edge angel is the angle between two lines from the center of the femoral head (stabled blue lines), respectively, a vertical line perpendicular to the reference line and a line connecting the center of the femoral head to the lateral sourcil of the acetabulum. The acetabular index angle is the angle between the horizontal reference line and a line (solid blue line) connecting the medial and lateral sourcils of the acetabulum. The foramen width is the length of a line across the widest part of the obturator foramen (Yellow lines) parallel to the horizontal reference line.

**Figure 3 diagnostics-12-02597-f003:**
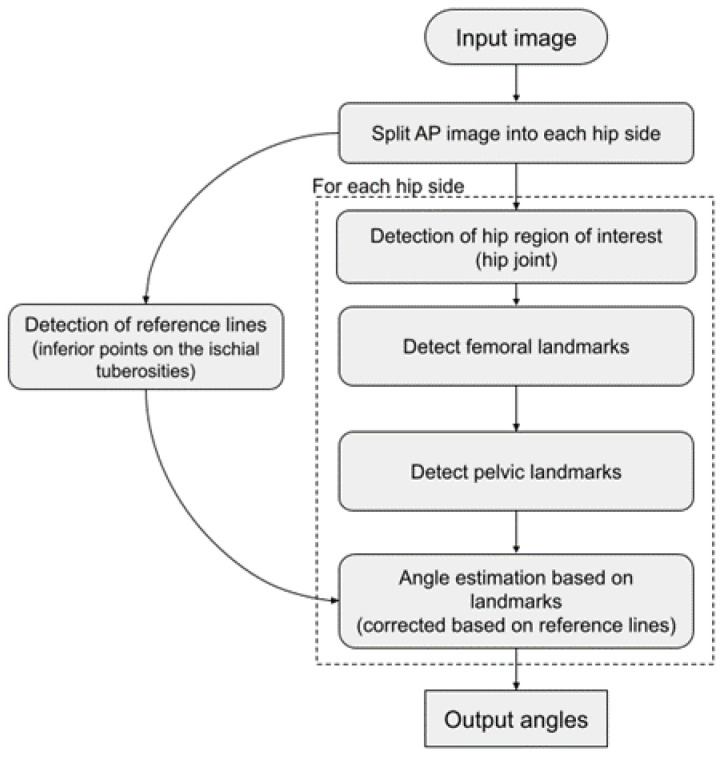
Algorithm flowchart.

**Figure 4 diagnostics-12-02597-f004:**
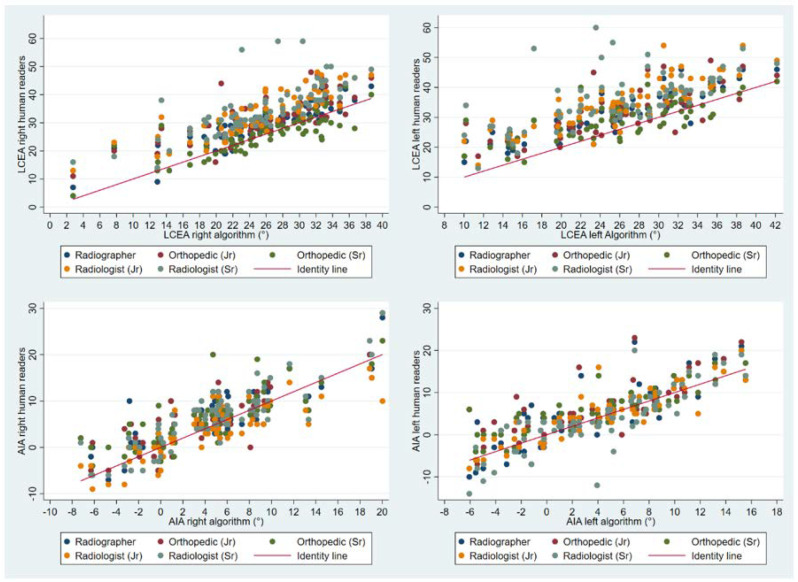
Scatter plots with human measurements of LCEA and AIA over algorithm measurements. LCEA; Lateral center edge angle, AIA; Acetabular index angle. Scatter plots in the top row line (LCEA) depict a tendency that human readers measure the LCEA at a higher value than the algorithm, although the senior orthopedic surgeon agree the most with the algorithm (green dots seen in close proximity to the red identity line).

**Figure 5 diagnostics-12-02597-f005:**
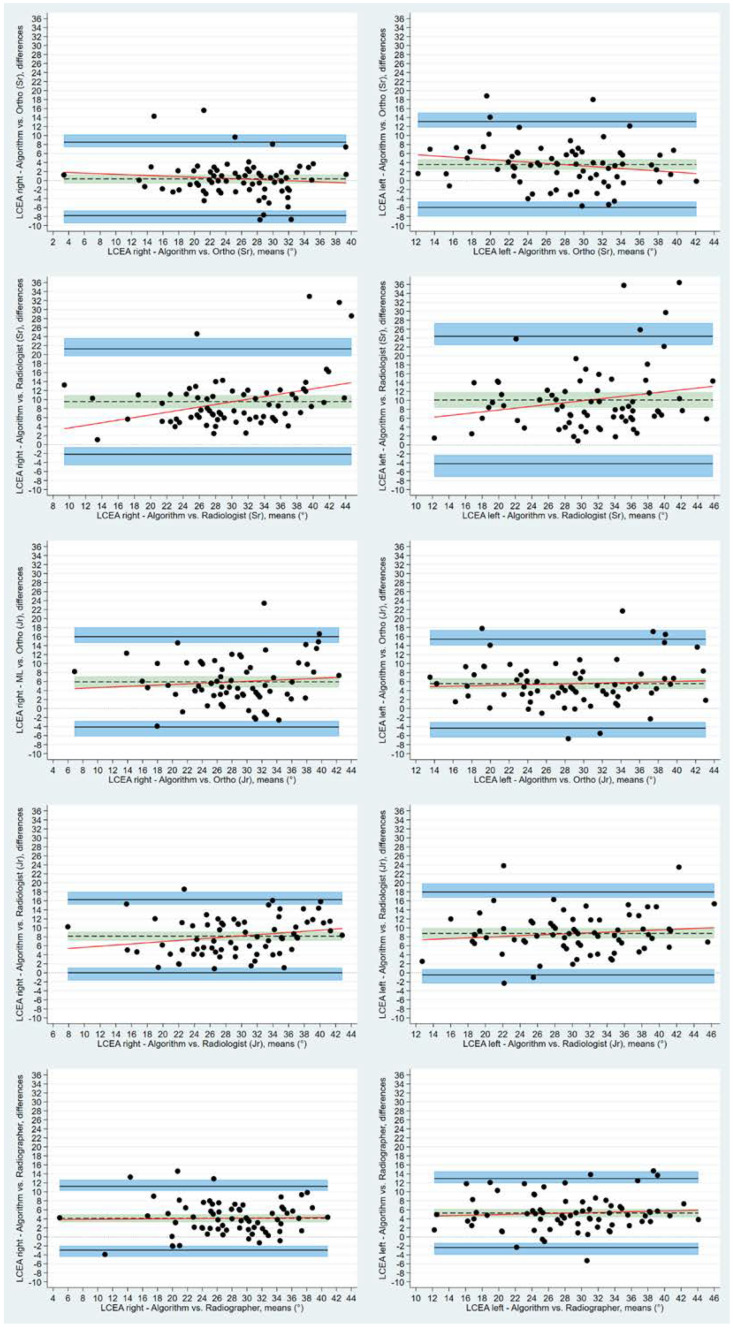
Bland-Altman plots with Limits of Agreement and respective 95% CI demonstrating agreement between individual human readers and algorithm for lateral center edge angles. Differences between measurements are plotted against the mean of the measurements. Solid black lines depict upper and lower limits of agreement. Shaded blue areas depict the 95% confidence intervals. Dotted black lines are the mean measured differences with the shaded green areas illustrating the 95% confidence intervals.

**Figure 6 diagnostics-12-02597-f006:**
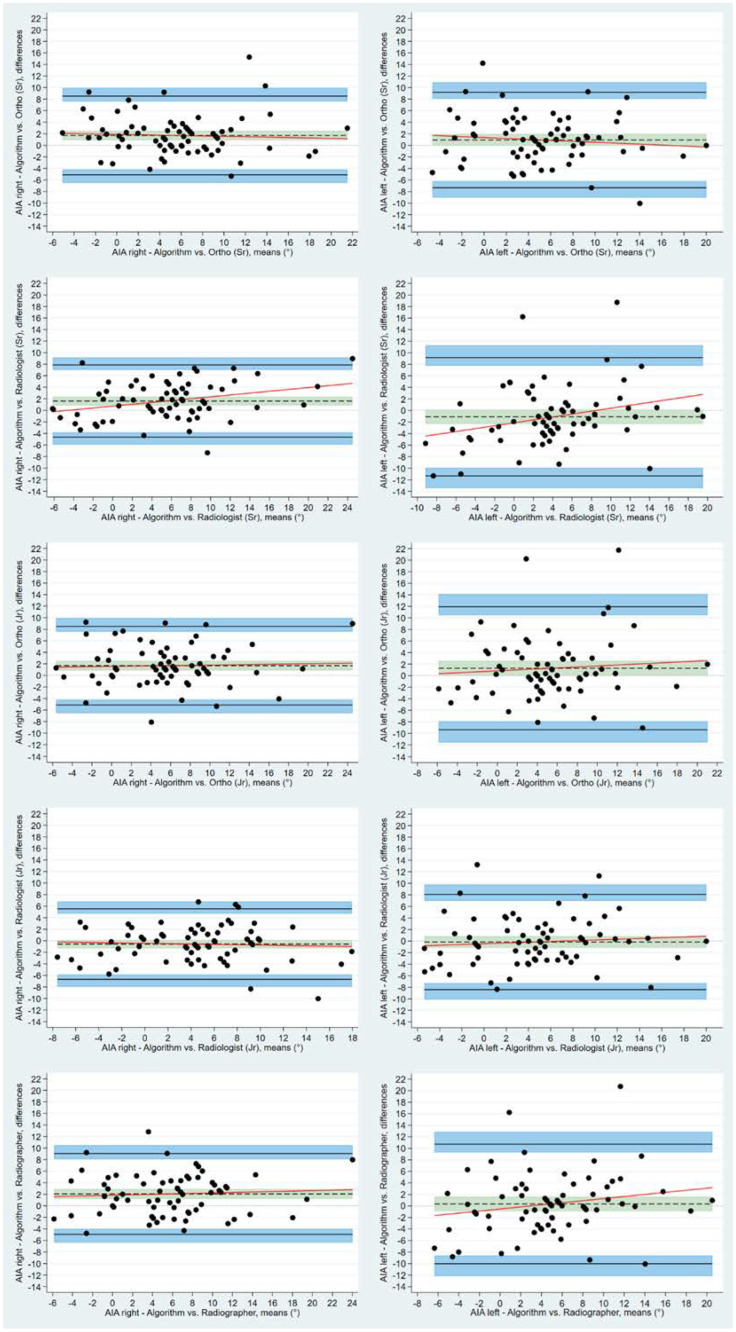
Bland-Altman plots with Limits of Agreement and respective 95% CI depicting agreement between human readers and algorithm for acetabular index angle. Differences between measurements are plotted against the mean of the measurements. Solid black lines depict upper and lower limits of agreement. Shaded blue areas depict the 95% confidence intervals. Dotted black lines are the mean measured differences with the shaded green areas illustrating the 95% confidence intervals.

**Table 1 diagnostics-12-02597-t001:** First read by algorithm (mean, SD, range, 1st and 3rd percentiles) and differences (Diff) between 1st and 2nd algorithm read. (n = 71).

	Mean (SD)	Mean (SD) ^Diff^	Range [min; ma×]	Range ^Diff^ [min; ma×]	Q_1_	Q_1_^Diff^	Q_3_	Q_3_^Diff^
**LCEA^Right^**	25.43 (6.96)	3.97 × 10^−14^ (4.9 × 10^−14^)	[2.77; 38.65]	[0; 0]	21.10	0	31.16	9.95 × 10^−14^
**LCEA^Left^**	25.91 (7.51)	4.78 × 10^−14^ (5.03 × 10^−14^)	[10.03; 42.16]	[0; 1.03 × 10^−12^]	20.84	0	31.33	9.95 × 10^−14^
**AIA^Right^**	4.69 (5.67)	4.8 × 10^−15^ (1.72 × 10^−14^)	[−7.22; 20.03]	[−1.02 × 10^−14^; 9.95 × 10^−14^]	0.23	0	8.29	9.77 × 10^−15^
**AIA^Left^**	4.03 (5.40)	8.14 × 10^−15^ (2.58 × 10^−14^)	[−6.06; 15.55]	[−1.02 × 10^−14^; −9.95 × 10^−14^]	−0.23	0	8.11	8.88 × 10^−14^

SD; Standard deviation, Diff; difference between 1st and 2nd read, min, minimum, max; maximum, Q_1_; 1st percentile, Q_3_; 3rd percentile, LCEA; lateral center edge angle, AIA; acetabular index angle.

**Table 2 diagnostics-12-02597-t002:** Mean measurements of LCEA and AIA including SD and range for all readers and Algorithm (A) (n = 71).

	LCEA (SD) [Range]		AIA (SD) [Range]	
	Right	Left	Right	Left
O^1^	29.5 (7.0) [7 to 43]	31.2 (7.8) [11 to 48]	6.7 (5.9) [−7 to 28]	5.0 (6.6) [−10 to 22]
O^2^	31.4 (7.4) [11 to 48]	31.4 (7.8) [17 to 49]	6.4 (5.8) [−5 to 29]	6.0 (6.1) [−7 to 23]
O^3^	25.8 (6.6) [4 to 43]	29.5 (6.6) [13 to 43]	6.4 (5.5) [−4 to 23]	5.6 (5.3) [−7 to 20]
O^4^	33.6 (7.8) [13 to 48]	34.7 (8.1) [14 to 54]	4.1 (5.5) [−9 to 17]	4.5 (6.0) [−8 to 20]
O^5^	35.0 (9.0) [14 to 59]	36.0 (8.9) [13 to 60]	6.3 (6.6) [−6 to 29]	3.6 (7.0) [−14 to 20]
A	25.4 (7.0) [3 to 39]	25.9 (7.5) [10 to 42]	4.7 (5.7) [−7 to 20]	4.0 (5.4) [−6 to 16]

LCEA; Lateral center edge angle, AIA; Acetabular index angle, SD; standard deviation, O^1^; reporting radiographer, O^2^; Orthopedic surgeon < 5 years of experience, O^3^; Orthopedic surgeon > 5 years of experience, O^4^; Radiologist < 5 years of experience, O^5^; Radiologist > 5 years of experience.

**Table 3 diagnostics-12-02597-t003:** Lateral center edge angle. Bland Altman limits of agreement and bias (mean and SD). Agreement between algorithm and individual human readers. (n = 71).

		Bias Mean (SD)	Bias 95% CI	Limits of Agreement	Lower Limit of Agreement 95% CI	Upper Limit of Agreement 95% CI
LCEA^right^	O^1^	4.13 (3.62)	3.28 to 4.99	−2.95 to 11.22	−4.43 to −1.99	10.25 to 12.69
	O^2^	5.93 (5.12)	4.72 to 7.15	−4.10 to 15.97	−6.19 to −2.73	14.60 to 18.05
	O^3^	0.37 (4.16)	−0.61 to 1.36	−7.79 to 8.53	−9.48 to −6.67	7.42 to 10.23
	O^4^	8.15 (4.14)	7.17 to 9.13	0.02 to 16.27	−1.66 to 1.13	15.16 to 17.96
	O^5^	9.56 (5.98)	8.14 to 10.97	−2.16 to 21.27	−4.59 to −0.56	19.67 to 23.70
LCEA^left^	O^1^	5.29 (3.91)	4.37 to 6.22	−2.37 to 12.95	−3.96 to −1.32	11.90 to 14.54
	O^2^	5.53 (5.04)	4.34 to 6.73	−4.36 to 15.42	−6.41 to −3.01	14.07 to 17.47
	O^3^	3.56 (4.87)	2.41 to 4.74	−5.99 to 13.10	−7.97 to −4.68	11.80 to 15.09
	O^4^	8.76 (4.70)	7.65 to 9.87	−0.45 to 17.96	−2.36 to 0.81	16.70 to 19.87
	O^5^	10.01 (7.29)	8.37 to 11.82	−4.19 to 24.38	−7.16 to −2.24	22.43 to 27.35

LCEA; Lateral center edge angle, SD; Standard deviation, CI; confidence interval, O^1^; Reporting Radiographer > 5 years of experience, O^2^; Orthopedic surgeon < 5 years of experience, O^3^; Orthopedic surgeon > 5 years of experience, O^4^; Radiologist < 5 years of experience, O^5^; Radiologist > 5 years of experience.

**Table 4 diagnostics-12-02597-t004:** Acetabular index angle. Bland Altman limits of agreement and bias (mean and SD). Agreement between algorithm and individual human readers. (n = 71).

		Bias Mean (SD)	Bias 95% CI	Limits of Agreement	Lower Limit of Agreement 95% CI	Upper Limit of Agreement 95% CI
AIA^right^	O^1^	2.06 (3.57)	1.21 to 2.90	−4.94 to 9.05	−6.39 to −3.98	8.10 to 10.50
	O^2^	1.69 (3.48)	0.87 to 2.51	−5.13 to 8.51	−6.55 to −4.20	7.58 to 9.93
	O^3^	1.70 (3.48)	0.88 to 2.53	−5.11 to 8.53	−6.54 to −4.19	7.59 to 9.94
	O^4^	−0.58 (3.12)	−1.32 to 0.16	−6.69 to 5.5	−7.96 to −5.86	4.70 to 6.80
	O^5^	1.62 (3.19)	0.86 to 2.37	−4.63 to 7.87	−5.93 to −3.78	7.02 to 9.17
AIA^left^	O^1^	0.35 (5.29)	−0.90 to 1.60	−10.01 to 10.72	−12.17 to −8.60	9.30 to 12.87
	O^2^	1.28 (5.44)	−0.01 to 2.57	−9.37 to 11.93	−11.58 to −7.92	10.48 to 14.15
	O^3^	0.93 (4.22)	−0.07 to 1.93	−7.34 to 9.19	−9.05 to −6.21	8.06 to 10.91
	O^4^	−0.17 (4.21)	−1.17 to 0.83	−8.42 to 8.08	−10.13 to −7.29	6.95 to 9.79
	O^5^	−1.09 (5.22)	−2.32 to 0.15	−11.32 to 9.15	−13.45 to −9.92	7.75 to 11.28

AIA, Acetabular index angle, SD Standard deviation, CI confidence interval, O^1^ Reporting Radiographer > 5 years of experience, O^2^ Orthopedic surgeon < 5 years of experience, O^3^ Orthopedic surgeon > 5 years of experience, O^4^ Radiologist < 5 years of experience, O^5^ Radiologist > 5 years of experience.

**Table 5 diagnostics-12-02597-t005:** Estimated variances derived from the linear effects model. Five human readers, LCEA and AIA. (n = 78).

	Component	Estimate	95 % CI	*p*-Value
**LCEA^Right^**				
	Constant	26.03	19.25 to 32.82	<0.0001
	Patient variance	39.46	27.80 to 56.01	
	Reader variance	11.88	2.9 to 48.62	
	Repeated measure variance	7.44	0.87 to 63.49	
	Residual variance	17.80	15.36 to 20.62	
**LCEA^Left^**				
	Constant	28.55	21.70 to 35.39	<0.0001
	Patient variance	46.32	32.77 to 65.49	
	Reader variance	7.73	1.86 to 32.14	
	Repeated measure variance	7.16	0.82 to 62.39	
	Residual variance	19.08	16.47 to 22.10	
**AIA^Right^**				
	Constant	4.84	0.66 to 9.02	0.023
	Patient variance	25.36	17.97 to 35.79	
	Reader variance	1.27	0.29 to 5.70	
	Repeated measure variance	1.25	N/A	
	Residual variance	8.93	7.71 to 10.34	
**AIA^Left^**				
	Constant	3.27	−1.40 to 7.93	0.170
	Patient variance	33.39	23.82 to 46.82	
	Reader variance	0.65	0.13 to 3.23	
	Repeated measure variance	4.89	N/A	
	Residual variance	8.80	7.60 to 10.20	
	Residual variance	111.37	96.18 to 128.97	

LCEA; Lateral center edge angle, AIA; Acetabular index angle, SD; standard deviation, N/A; Not available.

**Table 6 diagnostics-12-02597-t006:** Estimated repeatability coefficients for between and within reader measurements based on the linear effects model. Five human readers, LCEA and AIA. (n = 78).

	Repeatability Coefficient	
	Same Patient, Same Reader	Same Patient, Different Reader
LCEA^Right^	11.69	15.09
LCEA^Left^	12.10	14.34
AIA^Right^	8.28	8.85
AIA^Left^	8.22	8.52

LCEA; Lateral center edge angle, AIA; Acetabular index angle.

## Data Availability

Data from this study were not provided in any public places. The reader and algorithm measurements were stored in the secure RedCap database.

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
