# Peer review of "A Deep Learning Algorithm for Radiographic Measurements of the Hip in Adults—A Reliability and Agreement Study"

_diagnostics, 2022, doi:10.3390/diagnostics12112597_

Round 1

Reviewer 1 Report

A deep learning model to diagnose hip dysplasia on radiographs

Title is ok

The abstract is fine.

Introduction.

Can you define the age group of younger patients? 

Methods

How many patients did you exclude – consider including a figure for visualization

Please be consistent – readers or observers

Discussion

Perhaps a few words about clinical impact or value?

Overall, an interesting paper with a focus on comparing a model to clinicians.

Author Response

Reviewer 1

Dear Reviewer

Thank you very much for your constructive comments on our manuscript. We have addressed each of your concerns point by point. 

Introduction.

Can you define the age group of younger patients? 

I can see how younger may be conceived as pediatric, I have therefore added skeletally mature to the sentence. Hope this clarifies the issue.

Methods

How many patients did you exclude – consider including a figure for visualization

A figure, Figure 1, has been included on the exclusion/inclusion process

Please be consistent – readers or observers

Thank you for pointing this out. It has been corrected throughout.

Discussion

Perhaps a few words about clinical impact or value?

We agree, clinical value is important. In the last section of the discussion we therefore state:

”We predict this algorithm as one of many valuable tools in the future with the potential to improve measurement consistency. Potentially, with further development, this algorithm can act as a screening tool within departments of radiology and orthopedic surgery, minimizing delayed diagnosis by identifying abnormal findings on pelvic radiographs suspicious of dysplasia. Considering the vast rising bundle of medical imaging, tools to assist physicians and reporting radiographers are needed and integrating locally validated algorithms into clinical practice is essential.”

Reviewer 2 Report

Dear author,

Thank you very much for your great work. The study is interesting. But there are some points need to be explained in more detail. These are listed below.

1- How did you select the population? Are these radiographs belongs to normal people or patients who have hip dysplasia?

2-In my opinion, the demographics of the x rays can be given in a separate table.

3-The difference between the human measurements is interesting. Can you explain this situation in detail? 

4-The comment as " AI is more sensitive to detect dysplasia" is not possible. You have to give the normal values of angles to the system and the ones which are not in normal values should be detected as abnormal. So, if the system measures big values then humans, that means there is a problem about machine learning. 

5-Age of the patients is important. As over 50 the treatment for this situation is mainly total hip arthroplasty. For periacetabular osteotomy, the young population is the target. So it is better to give  the patients demographics.

6-One of the point is all the x rays were taken while weight bearing. Is it important to take them in 15 degrees internal rotation to see the exact femoral head?

Kind regards.

Author Response

Reviewer 2

Dear Reviewer

Thank you very much for your constructive comments on our manuscript. We have addressed each of your concerns point by point.

1- How did you select the population? Are these radiographs belongs to normal people or patients who have hip dysplasia?

Since this was the initial test of the algorithm in a clinical setting, we did not stratify based on the presence of dysplasia. Pelvic radiographs of adults referred to diagnostic workup in relation to non-traumatic hip pain at our hospital were consecutively included (Section 2.1). You are correct though, that this is important and the next step with the algorithm is indeed, to establish if the algorithm can differentiate between dysplasia vs. normal.

2-In my opinion, the demographics of the x rays can be given in a separate table.

We did not have permission to read patient records, hence the only patient information we have is age and sex. We therefore chose to present that data in the text as opposed to in a table (section 3).

3-The difference between the human measurements is interesting. Can you explain this situation in detail? 

There was indeed reader variance between the humans in our study. We have further elaborated on that in the discussion.

“High measurement variability between human readers has previously been reported in regard to radiographic measurements used to assess hip dysplasia. Although the human readers in the current study were presented with a protocol defining the measurements, the repeatability coefficient was high from a clinical perspective, i.e. ranging from 12 to 15° for LCEA measurements. Perhaps a consensus meeting with all human readers, prior to the measuring sessions could have clarified definition of landmarks further; particularly the extent of the lateral sourcil may have affected inter-observer variance. Moreover, the human readers electronically drew and manually positioned a best-fit circle for the femoral head in the PACS which is also an inherent limitation that may have contributed to inconsistency in LCEA measurements. Although the BA LoA for the right LCEA was narrow for reader 3, no other systematic differences in measurement variability were uncovered. Hence, educational background or experience cannot explain variation in data.”

4-The comment as "AI is more sensitive to detect dysplasia" is not possible. You have to give the normal values of angles to the system and the ones which are not in normal values should be detected as abnormal. So, if the system measures big values then humans, that means there is a problem about machine learning. 

We agree that the sentence “AI is more sensitive to detect dysplasia” is not sensible. The current model does not diagnose dysplasia, it makes the measurements and the physician can use the measurements as a support tool in the clinical setting.

5-Age of the patients is important. As over 50 the treatment for this situation is mainly total hip arthroplasty. For periacetabular osteotomy, the young population is the target. So it is better to give the patients demographics.

 We agree, for PAO, it is mainly the younger population. Since this was the initial test of the algorithm, we chose to include younger and older patients with an average age of 50.1 years [range; 18 to 91] consisting of 36 females and 35 males (Section 3). The next step (new study) will be to explore if the algorithm can differentiate between dysplasia vs. normal hips in an appropriate age group, i.e. below 50.

6-One of the point is all the x rays were taken while weight bearing. Is it important to take them in 15 degrees internal rotation to see the exact femoral head?

 So true, internal rotation of the legs are important. Thank you for pointing this out. Weight bearing pelvic X-rays are taken with legs internally rotated 15 degrees at our institution. This has now been added in section 2.2.

Reviewer 3 Report

1.     It is recommended that a review of the relevant research background be added to the introduction section of section 1 of the article.

2.     The algorithm introduction in Section 2.4 is relatively brief, a detailed description is recommended, experiment-specific strategies are recommended for additional instructions.

3.     It is recommended that Section 2.4 add an algorithm model flowchart.

4.     It is recommended to add comparative experiments with other algorithms such as ResNet50 after section 2.4 to increase the richness of the article and objectively prove the advantages of the proposed algorithm.

5.     The application software introduction section of the algorithm in section 2.5 of Recommendation is clearly displayed in the relevant flowcharts and software interfaces.

6.     It is recommended to add a comparison of the results of the state the art methods with the results of the state the art methods in the third section to prove the reliability of this paper, and the following examples are given in the literature.

A. R. Hareendranathan et al., "Toward automatic diagnosis of hip dysplasia from 2D ultrasound," 2017 IEEE 14th International Symposium on Biomedical Imaging (ISBI 2017), 2017, pp. 982-985, doi: 10.1109/ISBI.2017.7950680.

Author Response

Reviewer 3

Dear Reviewer

Thank you very much for your constructive comments on our manuscript. We have addressed each of your concerns point by point.

  1. It is recommended that a review of the relevant research background be added to the introduction section of section 1 of the article.

A section in the introduction has been added on relevant research background.

  1. The algorithm introduction in Section 2.4 is relatively brief, a detailed description is recommended, experiment-specific strategies are recommended for additional instructions.

A detailed description of the development of the algorithm is in our opinion beyond the scope of this paper.  The purpose of this study was to do a clinical evaluation of a proprietary algorithm to assess how it performs against our clinical personnel. Our aim is to further evaluate the algorithm clinical partly retrospectively and prospectively in the next study.

  1. It is recommended that Section 2.4 add an algorithm model flowchart.

Thank you for this suggestion. An RBhip model flowchart has been added in section 2.4

  1. It is recommended to add comparative experiments with other algorithms such as ResNet50 after section 2.4 to increase the richness of the article and objectively prove the advantages of the proposed algorithm.

As for the response to reviewer item 2 above, the scope of this study was to do a clinical evaluation of  a proprietary algorithm. We agree that comparing model architecture, hyperparameter selection, and similar design choices are important for the development of DL-based systems. These tests have been performed by the company who developed the algorithm as part of their internal testing and verification. However, these experiments (which were done on a separate dataset) are in our opinion beyond the scope of this study. Additionally, the high inter-reader reliability and thus difficulties in establishing a reliable reference standard in our dataset, will make it difficult to repeat these experiments in a meaningful way in this study.

  1. The application software introduction section of the algorithm in section 2.5 of Recommendation is clearly displayed in the relevant flowcharts and software interfaces.

The algorithm work-flow is described on page 5 (line 191-4). Hope this clarifies your question.

  1. It is recommended to add a comparison of the results of the state the art methods with the results of the state the art methods in the third section to prove the reliability of this paper, and the following examples are given in the literature.

  1. R. Hareendranathan et al., "Toward automatic diagnosis of hip dysplasia from 2D ultrasound," 2017 IEEE 14th International Symposium on Biomedical Imaging (ISBI 2017), 2017, pp. 982-985, doi: 10.1109/ISBI.2017.7950680.

 Thanks for the addition, but the current national standard for diagnosis of dysplasia in adults is made by a complete medical history, a physical examination, and an X-ray evaluation. Sometimes other types of imaging. We have, however, added a section on AI and the diagnosis of hip dysplasia including ultrasound in the introduction.